# Do Deep Networks Transfer Invariances across Classes?

**Allan Zhou**[*] **& Fahim Tajwar**[*]
Stanford University

**Alexander Robey**
University of Pennsylvania

**Tom Knowles**
Stanford University

**George J. Pappas & Hamed Hassani**
University of Pennsylvania

**Chelsea Finn**
Stanford University

## Abstract

To generalize well, classifiers must learn to be invariant to nuisance transformations that do not alter an input's class. Many problems have "class-agnostic" nuisance transformations that apply similarly to all classes, such as lighting and background changes for image classification. Neural networks can learn these invariances given sufficient data, but many real-world datasets are heavily class imbalanced and contain only a few examples for most of the classes. We therefore pose the question: how well do neural networks transfer class-agnostic invariances learned from the large classes to the small ones? Through careful experimentation, we observe that invariance to class-agnostic transformations is still heavily dependent on class size, with the networks being much less invariant on smaller classes. This result holds even when using data balancing techniques, and suggests poor invariance transfer across classes. Our results provide one explanation for why classifiers generalize poorly on unbalanced and long-tailed distributions. Based on this analysis, we show how a generative approach for learning the nuisance transformations can help transfer invariances across classes and improve performance on a set of imbalanced image classification benchmarks. Source code for our experiments is available at `https://github.com/AllanYangZhou/generative-invariance-transfer`.

## 1 Introduction

Good generalization in machine learning models requires ignoring irrelevant details: a classifier should respond to whether the subject of an image is a cat or a dog but not to the background or lighting conditions. Put another way, generalization involves *invariance* to nuisance transformations that should not affect the predicted output. Deep neural networks can and do learn invariances given sufficiently many diverse examples. For example, we can expect our trained "cat vs dog" classifier to be invariant to changes in the background provided the training dataset contains images of cats and dogs with varied scenery. However, if all the training examples for the "dog" class are set in grassy fields, our classifier might be confused by an image of a dog in a house (Beery et al., 2018).

This situation is problematic for *imbalanced* datasets, in which the amount of training data varies from class to class. Class imbalance is common in practice, as many real-world datasets follow a long-tailed distribution where a few *head* classes have many examples, and each of the remaining *tail* classes have few examples. Hence even if the cumulative number of examples in a long-tailed dataset is large, classifiers may struggle to learn invariances for the smaller tail classes. And while augmentation can address this issue by increasing the amount and variety of data in the tail classes, this strategy is not feasible for every nuisance transformation (e.g., modifying an image's background scenery). On the other hand, many nuisance transformations such as lighting changes are "class agnostic:" they apply similarly to examples from any class, and should generalize well across classes (Hariharan & Girshick, 2016). Ideally a trained model should automatically transfer invariance to class agnostic transformations from larger classes to smaller ones. This observation raises the question: *how well do deep neural network classifiers transfer learned invariances across classes*?

---

[*]First two authors contributed equally.

Expected KL divergence (eKLD) under transformation

Figure 1: The expected KL divergence (eKLD, Eq. 2) measures how the classifier's estimated label probabilities $\hat{P}_w(\cdot|x)$ change under nuisance transformation, with **lower eKLD corresponding to more invariance**. Each plot shows per-class eKLD of classifiers trained on a long-tailed Kuzushiji-49 (K49) dataset variant, arranged by class size. Classifiers trained by either empirical risk minimization (ERM) or delayed resampling (CE+DRS) show the same trend: invariance depends heavily on class size, and classifiers are more invariant on images from the larger classes. Shaded regions show 95% CIs over 30 trained classifiers.

In this work, we find empirical evidence that neural networks transfer learned invariances poorly across classes, even *after* applying balancing strategies like oversampling. For example, on a long-tailed dataset where every example is rotated uniformly at random, classifiers tend to be rotationally invariant for images from the head classes but not on images from the tail classes. To this end, we present a straightforward method for more effectively transferring invariances across classes. We first train an input conditioned but class-agnostic generative model which captures a dataset's nuisance transformations, where withholding explicit class information encourages transfer between classes. We then we use this generative model to transform training inputs, akin to learned data augmentation training the classifier. We ultimately show that the resulting classifier is more invariant to the nuisance transformations due to better invariance on the tail classes, which in turn leads to better test accuracy on those classes.

**Contributions.** The primary contribution of this paper is an empirical study concerning whether deep neural networks learn invariances in long-tailed and class-imbalanced settings. We analyze the extent to which neural networks (fail to) transfer learned invariances across classes, and we argue that this lack of invariance transfer partially explains poor performance on real-world class-imbalanced datasets. Our analysis suggests that one path to improving imbalanced classification is to develop approaches that better transfer invariances across classes. We experiment with one such approach, which we call Generative Invariance Transfer (GIT), in which we train a generative model of a task's nuisance transformations and then use this model to perform data augmentation of small classes. We find that combining GIT with existing methods such as resampling improves balanced accuracy on imbalanced image classification benchmarks such as GTSRB and CIFAR-LT.

## 2 RELATED WORK

Real-world datasets are often class imbalanced, and state-of-the-art classifiers often perform poorly under such conditions. One particular setting of interest is when the class distribution is *long-tailed*, where most of the classes have only a few examples (Liu et al., 2019; Guo et al., 2016; Thomee et al., 2016; Horn & Perona, 2017). Researchers have proposed many approaches for this setting, including correcting the imbalance when sampling data (Buda et al., 2018; Huang et al., 2016; Chawla et al., 2002; He & Garcia, 2009), using modified loss functions (Cao et al., 2019; Lin et al., 2017; Cui et al., 2019), and modifying the optimizer (Tang et al., 2020). These methods generally do not study the question of invariance transfer and are complementary to Generative Invariance Transfer. Yang & Xu (2020) study the mixed value of imbalanced labels and propose learning class-agnostic information using self-supervised training, while GIT learns class-agnostic transformations to use as data augmentation. Wang et al. (2017) propose implicit transfer learning from head to tail classes using a meta-network that generates the weights of the classifier's model. Here we quantitatively measure per-class invariances to more precisely explain why transfer learning may help on imbalanced problems, and *explicitly* transfer across classes with a generative model.

Toward understanding the failure modes of classifiers trained on real-world datasets, a rapidly growing body of work has sought to study the robustness of commonly-used machine learning models.

Notably, researchers have shown that the performance state-of-the-art models is susceptible to a wide range of adversarial attacks (Biggio & Roli, 2018; Goodfellow et al., 2014; Madry et al., 2017; Wong & Kolter, 2018; Robey et al., 2021a) and distributional shifts (Taori et al., 2020; Hendrycks et al., 2020; Koh et al., 2021). Ultimately, the fragility of these models is a significant barrier to deployment in safety-critical applications such as medical imaging (Bashyam et al., 2020) and autonomous driving (Chernikova et al., 2019). In response to these fragilities, recent methods have learned the distributional shifts present in the data using generative models and then use these learned transformations for robust training (Robey et al., 2020; 2021b; Wong & Kolter, 2020). These works assume pairs or groupings of transformed and untransformed examples, so that a generative model can learn the distribution shift. In our imbalanced setting, generative invariance transfer aims to learn transformations from the head classes that apply to the tail classes of the *same* dataset, and does not assume the data is paired or grouped.

There is a general interest in obtaining invariances for machine learning models (Benton et al., 2020; Zhou et al., 2021). Data augmentation (Beymer & Poggio, 1995; Niyogi et al., 1998) can be used to train classifiers to be invariant to certain hand-picked transformations, but requires the practitioner to know and implement those transformations in advance. In search of greater generality, Antoniou et al. (2017) use a trained GAN as data augmentations for training a downstream classifier. Similarly, Mariani et al. (2018) use a GAN to generate more examples for the minority classes in imbalanced classification. Learned augmentation can also be done in feature space, to avoid learning a generative model of the high dimensional input space (Yin et al., 2018; Chu et al., 2020). In the low-shot setting Hariharan & Girshick (2016) study the notion of transferring generalizable (or class-agnostic) variation to new classes, and Wang et al. (2018) learn to produce examples for new classes using meta-learning. Work in neural style transfer (Gatys et al., 2015b;a; Johnson et al., 2016) has long studied how to leverage trained neural networks to transfer variation between images or domains. In particular, GIT leverages advances in image-to-image translation (Isola et al., 2016; Zhu et al., 2017; Huang et al., 2018) as a convenient way to learn nuisance transformations and transfer them between classes. Our work carefully quantifies invariance learning under class imbalance, which can explain *why* leveraging generative models of transformations can improve performance. There is prior work analyzing how learned representations and invariances are affected by noise and regularization (Achille & Soatto, 2018), task diversity (Yang et al., 2019), and data diversity (Madan et al., 2021). The latter is most relevant to our own analysis, which studies how invariance is affected by the amount of data *per class*.

## 3 MEASURING INVARIANCE TRANSFER IN CLASS-IMBALANCED DATASETS

In this section we empirically analyze the invariance of trained classifiers to nuisance transformations, and the extent to which these classifiers transfer invariances across classes. In particular, we first introduce concepts related to invariance in the imbalanced setting, then define a metric for measuring invariance before describing our experimental setup. Finally, we present and analyze the observed relationship between invariance and class size.

### 3.1 SETUP: CLASSIFICATION, IMBALANCE, AND INVARIANCES

In the classification setting, we have input-label pairs $(x, y)$, with $y$ taking values in $\{1, \cdots, C\}$ where $C$ is the number of classes. We will consider a neural network model parameterized by weights $w$ that is trained to estimate the conditional probabilities $\hat{P}_w(y = j|x)$. The classifier selects the class $j$ with the highest estimated probability. Given a training dataset $\{(x^{(i)}, y^{(i)})\}_{i=1}^N \sim \mathbb{P}_{\text{train}}$, empirical risk minimization (ERM) minimizes average loss over training examples. But in the *class-imbalanced* setting, the distribution of $\{y^{(i)}\}$ in our training dataset is not uniform, and ERM tends to perform poorly on the minority classes. In real world scenarios we typically want to perform well on all classes, e.g. in order to classify rare diseases properly (Bajwa et al., 2020) or to ensure fairness in decision making (Hinnefeld et al., 2018). Hence we evaluate classifiers using *class-balanced* metrics, which is equivalent to evaluating on a test distribution $\mathbb{P}_{\text{test}}$ that *is* uniform over $y$.

To analyze invariance, we assume there is a distribution $T(\cdot|x)$ over nuisance transformations of $x$. Given that nuisance transformations do not impact the labels, we expect a good classifier to be *invariant* to such transformations, i.e.,

$$\hat{P}_w(\cdot|x) = \hat{P}_w(\cdot|x'), \quad x' \sim T(\cdot|x) \tag{1}$$

That is, the estimated conditional class probabilities should be unchanged by these transformations.

### 3.2 Measuring learned invariances

To quantify the extent to which classifiers learn invariances, we measure the expected KL divergence (eKLD) between its estimated class probabilities for original and transformed inputs:

$$\text{eKLD}(\hat{P}_w) = \mathbb{E}_{x \sim \mathbb{P}_{\text{train}}, x' \sim T(\cdot|x)} \left[ D_{\text{KL}} \left( \hat{P}_w(\cdot|x) || \hat{P}_w(\cdot|x') \right) \right] \quad (2)$$

This is a non-negative number, with **lower eKLD corresponding to more invariance**; a classifier totally invariant to $T$ would have an eKLD of 0. We can estimate this metric in practice given a trained classifier if we have a way to sample $x' \sim T(\cdot|x)$. To study how invariance depends on class size, we can also naturally compute the *class-conditional* eKLD by restricting the expectation over $x$ to examples from a given class $j$.

Calculating eKLD and studying invariance requires access to a dataset's underlying nuisance transformation distribution $T$, but for most real world datasets we do not know $T$ a-priori. Instead, we create synthetic datasets using a chosen nuisance distribution. Like RotMNIST (Larochelle et al., 2007), a common benchmark for rotation invariance, we create these datasets by transforming each example from a natural dataset. This is different from data augmentation, where *multiple* randomly sampled transformations would be applied to the same image throughout training. We want to test how classifiers learn invariances from a limited amount of data diversity; providing the transformation as data augmentation artificially boosts the observed data diversity and we use it as an Oracle comparison in later experiments.

We modify Kuzushiji-49 (Clanuwat et al., 2018) to create three synthetic datasets using three different nuisance transformations: image rotation (K49-ROT-LT), varying background intensity (K49-BG-LT), and image dilation or erosion (K49-DIL-LT). Appendix Figure 7 shows representative image samples from each of these datasets. To make the training datasets long-tailed (LT), we choose an arbitrary ordering of classes from largest to smallest. Then we selectively remove examples from classes until the frequency of classes in the training dataset follows Zipf's law with parameter $2.0$, while enforcing a minimum class size of $5$. We repeat this process with $30$ randomly sampled orderings of the classes from largest to smallest, to construct $30$ different long-tailed training datasets for each variant. Each long-tailed training set has $7864$ training examples, with the largest class having $4828$ and the smallest having $5$. Since we are interested in either per-class or class-balanced test metrics, we do not modify the test set class distribution.

Good classifiers for K49-ROT-LT, K49-BG-LT, and K49-DIL-LT should clearly be rotation, background, and dilation/erosion invariant, respectively, for inputs from any class. We can train classifiers and measure their invariance for inputs of each class by estimating per-class eKLD. That is, we first sample more transformations and then measure how predictions change on held-out test inputs. For training we use both standard **ERM** and **CE+DRS** (Cao et al., 2019), which stands for delayed (class-balanced) resampling with standard cross-entropy loss. DRS samples training examples naively just as in ERM for the initial epochs of training, then switches to class-balanced sampling for the later epochs of training. We train a separate classifier using each method on each training dataset, then calculate each classifier's per-class eKLD on held out test examples.

Figure 1 shows how the resulting per-class eKLD varies with class size. The eKLD curves for both methods show a clear pattern on all three transformation families: invariance decreases as classes get smaller. This result, while intuitive, shows quantitatively that deep neural networks learn invariances non-uniformly on imbalanced datasets. It suggests that they fail to transfer learned invariances across classes despite the nuisance transformation families being fairly class-agnostic. Although these results use a ResNet architecture, Appendix C.2 shows similar results for non-ResNet CNNs. Note that by construction, smaller classes contain both fewer original K49 examples and less transformation diversity. Yet Appendix 8 shows the same trend for datasets where the number of original K49 examples is the same across all classes, with larger classes only containing more sampled transformations of the same images. This shows that the trend is largely due to observed transformation diversity, rather than the number of original K49 examples.

Figure 1 also shows that training classifiers with resampling (CE+DRS) results in generally lower eKLD (more invariance) for the same class size. This effect may partly explain why resampling can improve class balanced metrics such as balanced test accuracy. However, the eKLD curves show that there is clearly still room for improvement and even DRS does not achieve close to uniform invariance learning across classes. One explanation for this is that DRS can show the classifier the

Figure 2: Samples from MIITNs trained to learn the nuisance transformations of K49-BG-LT (background intensity variation) and K49-DIL-LT (dilation/erosion). Each row shows multiple transformations of the same original image. We see a diversity of learned nuisances, even for inputs from the smallest class (bottom row).

same minority class images more often, but cannot increase the transformation diversity of those images (e.g., DRS cannot by itself create more examples of rotated images from the minority classes). This suggests that learning the transformation distribution, combined with resampling schemes, can help classifiers achieve uniform invariance across classes.

## 4 TRANSFERRING INVARIANCES WITH GENERATIVE MODELS

We've seen that classifiers do a poor job learning invariance to nuisance transformations on the tail classes of long-tailed datasets. Here we explain how generative invariance transfer (GIT) can transfer invariance across classes by *explicitly* learning the underlying nuisance distribution $T(\cdot|x)$.

### 4.1 LEARNING NUISANCE TRANSFORMATIONS FROM DATA

If we have the relevant nuisance transformations, we can use them as data augmentation to enforce invariance across all classes. Since this is rarely the case in practice, GIT approximates the nuisance distribution $T(\cdot|x)$ by training an *input-conditioned* generative model $\tilde{T}(\cdot|x)$. An input-conditioned generative model of the transformation is advantageous for class-agnostic nuisances. Since the class-specific features are already present in the input, the model can focus on learning the class-agnostic transformations, which by assumption should transfer well between classes.

To train the generative model in the image classification setting, we borrow architectures and training procedures from the literature concerning multimodal image-to-image translation networks (MI-ITNs). MIITNs can transform a given input image $x$ according to different nuisances learned from the data to generate varied output samples. The *multimodality* is ideal for capturing the full diversity of nuisance transformations present in the training data. Our experiments build off of a particular MIITN framework called MUNIT (Huang et al., 2018), which we modify to learn transformations between examples in a single dataset rather than between two different domains (see Appendix A for details). As we are not aware of any work that successfully trains MUNIT-like models to learn rotation, we focus on training MUNIT models to learn background intensity variation (K49-BG-LT) and dilation/erosion (K49-DIL-LT). Figure 2 shows the diversity of transformations these models produce from a given input. We see qualitatively that these models properly transform even inputs from the smallest class, evidencing successful transfer between classes. While these and later results show MUNIT successfully learns certain natural and synthetic transformations in imbalanced datasets, we note that GIT does not make MUNIT-specific assumptions, and other approaches for learning transformations can be considered depending on the setting.

Once we have the trained generative model, GIT uses the generative model as a proxy for the true nuisance transformations to perform data augmentation for the classifier, with the goal of improving invariance to these nuisance transformations for the small classes. Given a training minibatch $\{(x^{(i)}, y^{(i)})\}_{i=1}^{|B|}$, we sample a transformed input $\tilde{x}^{(i)} \leftarrow \tilde{T}(\cdot|x^{(i)})$ while keeping the label fixed. This augments the batch and boosts the diversity of examples that the classifier sees during training, particularly for the smaller classes. The pre-augmentation batches can be produced by an arbitrary sampling routine BATCHSAMPLER, such as a class-balanced sampler. We can also augment more selectively to mitigate the possibility of defects in the generative model hurting performance, especially on the large classes where we already observe good invariance and where the augmentation may be unnecessary. First, we introduce a cutoff $K$ such that GIT only augments examples from classes with fewer than $K$ examples. Second, we use GIT to augment only a proportion $p \in [0, 1]$ of each batch, so that the classifier sees a mix of augmented and "clean" data. Typically $p = 0.5$ in our experiments, and $K$ can range between $20 - 500$ depending on the dataset. Algorithm 1 details the GIT augmentation procedure explicitly.

---

**Algorithm 1** Generative Invariance Transfer: Classifier Training

---

**Input:** $\mathcal{D}$, the imbalanced training dataset
**Input:** BATCHSAMPLER, a minibatch sampling subroutine
**Input:** $\tilde{T}$, a generative model of transformations trained on $\mathcal{D}$
**Input:** UPDATEMODEL, a procedure that updates a classifier model given a minibatch
   **while** not done **do**
      $B \leftarrow \{(x^{(i)}, y^{(i)})\}_{i=1}^{|B|} = $ BATCHSAMPLER$(\mathcal{D})$               ▷ Sample raw batch
      **for** $i = 1, \cdots, $ Round$(p \times |B|)$ **do**            ▷ Augment proportion $p$ of batch
         **if** CLASSSIZE$(y^{(i)}) \leq K$ **then**        ▷ Optionally only augment the smaller classes
            Remove $(x^{(i)}, y^{(i)})$ from $B$
            Sample $\tilde{x}^{(i)} \leftarrow \tilde{T}(\cdot | x^{(i)})$           ▷ Transform input with generative model
            Add $(\tilde{x}^{(i)}, y^{(i)})$ to $B$
         **end if**
      **end for**
      UPDATEMODEL$(B)$                   ▷ Update model on batch
   **end while**

---

### 4.2 GIT IMPROVES INVARIANCE ON SMALLER CLASSES

As we saw in Figure 2, MIITNs can be trained to learn the relevant nuisance transformations for K49-BG-LT and K49-DIL-LT. Thus, given a trained MIITN, we aim to evaluate whether GIT can actually improve invariance on the smaller classes by transfering invariances from large to small classes. To do so, we measure the per-class eKLD metric on GIT-trained classifiers, with lower eKLD suggesting greater invariance. In particular, we train the classifiers using Algorithm 1 where BATCHSAMPLER is the delayed resampler and where UPDATE-CLASSIFIER uses gradient updates with respect to the cross-entropy (CE) loss function; we refer to this combined method as **CE+DRS+GIT (all classes)**. "All classes" refers to the fact that,

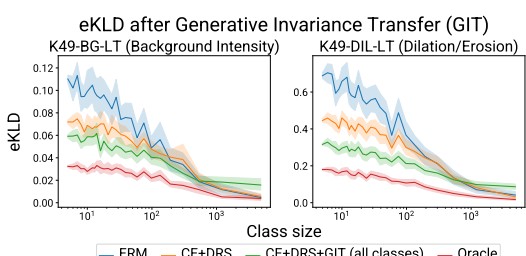

Figure 3: We observe that the expected KL divergence (eKLD) is lower for smaller classes when using generative invariance transfer (GIT). That is, GIT makes classifiers more uniformly invariant to the nuisance transform regardless of class size.

for the K49 experiments only, we disable the class size cutoff $K$, to observe GIT's effect at all class sizes. As an "Oracle" comparison we also replace GIT augmentation by the true nuisance transformation we used to construct the dataset, providing an unlimited amount of true transformation diversity (the training scheme is otherwise identical to CE+DRS).

Figure 3 shows the average per-class eKLD of classifiers trained by CE+DRS+GIT, compared with ERM, CE+DRS, and the Oracle. We clearly observe that GIT achieves more uniform invariance across class sizes than CE+DRS or ERM, with the biggest improvements for the smallest classes. We also observe that GIT can hurt invariance on the largest classes, where there is already sufficient ground truth data such that imperfections in the generative model end up degrading performance. This will justify using the class-size cutoff $K$ for GIT augmentations in our later experiments.

## 5 EXPERIMENTS

We have seen in Section 3 that classifiers transfer invariances poorly and can struggle to learn invariances on the smallest classes, and in Sec 4 that Generative Invariance Transfer (GIT) can lead to more uniform invariances across all class sizes. In this section, we verify that more invariant classifiers do indeed perform better, as measured by metrics such as balanced test accuracy. We also aim to understand whether GIT can be combined with other techniques for addressing class imbalance, and determine whether it is empirically important to only augment smaller classes.

### 5.1 EXPERIMENTAL SETUP

**Comparisons.** In our main comparisons, we will evaluate the performance of Generative Invariance Transfer (GIT) and study how it interacts with other techniques for addressing class imbalance,

which include resampling schemes and modified loss functions. We use **ERM** to denote the standard training procedure with cross entropy loss, where examples for training minibatches are sampled uniformly from the dataset. As in Section 3 we can alternatively train with delayed resampling, which we will refer to as **CE+DRS**. In addition to the typical cross entropy loss, we will compare to two recent loss functions that are representative of state-of-the-art techniques for learning from long-tailed data: the **LDAM** (Cao et al., 2019) and **Focal** (Lin et al., 2017) losses, which are designed to mitigate the imbalanced learning problem. Throughout the experiments, we will combine these loss functions with different sampling strategies (e.g. CE+DRS) and (optionally) GIT, e.g. "LDAM+DRS+GIT." For the K49 experiments in particular, we also compare against an **Oracle** that does data augmentation with the same transformation we used to construct the dataset. We evaluate all methods by balanced test set accuracy.

**Datasets.** We evaluate combinations of these methods on several long-tailed image classification benchmarks. Following the standard in long-tailed literature, we construct long-tailed training datasets by removing examples from certain classes until the class frequencies follow a long-tailed distribution. Appendix Figure 7 shows random samples from each the head and tail classes of each dataset. Aside from the long-tailed Kuzushiji-49 (*K49-LT*) variants of Section 3, we use:

*GTSRB-LT* is a long-tailed variant of GTSRB (Stallkamp et al., 2012; 2011) which contains images from 43 classes of German traffic signs in a large variety of lighting conditions and backgrounds, a natural source of nuisance transformations. We resize every image to have dimensions 32x32, randomly sample 25% of our training set as the validation set and use the rest to construct the long-tailed training dataset. We make the training dataset long-tailed by selectively removing examples so that the class frequencies follow a Zipf's law with parameter 1.8. We fix the minimum number of training samples for a particular class to be 5 — resulting in a training dataset where the most frequent class has 1907 examples, and the least frequent class has 5 examples. Since we are calculating class-balanced test metrics, we leave the test set unchanged.

*CIFAR-10-LT and CIFAR-100-LT* are long-tailed CIFAR (Krizhevsky, 2009) variants used in previous long-tailed classification literature (Cao et al., 2019; Tang et al., 2020). It has with $32 \times 32$ images in 10 or 100 class, respectively. Our setup is identical to Cao et al. (2019), with class frequency in the training set following an exponential distribution with the imbalance ratio (ratio of number of training examples between most frequent and least frequent class) set to 100. Similar to GTSRB-LT, we keep the test sets unchanged as we are calculating class-balanced test metrics.

*TinyImageNet-LT* is constructed from TinyImageNet (Le & Yang, 2015) similarly to CIFAR-LT. TinyImageNet has 200 classes, with 500 training and 50 test example per class. We remove training examples to achieve an imbalance ratio of 100 and keep the test sets unchanged.

*iNaturalist* is a large scale species detection dataset (Horn et al., 2018), with 8142 classes, 437,513 training and 24,426 validation images. The training set is naturally long-tailed and the validation set is balanced.

**Training.** For GTSRB-LT and CIFAR-LT we train ResNet32 (He et al., 2015) models for 200 epochs with batch size 128, optimized by SGD with momentum 0.9, weight decay $2 \times 10^{-4}$, and initial learning rate 0.1. The learning rate decays by a factor of 10 at epochs 160 and 180. For K49-LT we use a ResNet20 backbone trained for 50 epochs, with learning rate decays at 30 and 40 epochs. For TinyImageNet-LT we use an EfficientNet-b4 (Tan & Le, 2019) backbone and a cosine annealing learning rate scheduler (Loshchilov & Hutter, 2017). For iNaturalist-2018, we train a ResNet50 backbone for 90 epochs with a learning rate 0.1, with the learning rate annealed by 0.01 and 0.001 at epochs 30 and 60. See appendices B and C.1 for further classifier training details. We implement the GIT MIITNs using MUNIT (Huang et al., 2018) trained for $140,000$ steps (GTSRB-LT and CIFAR-LT), $200,000$ steps (TinyImageNet-LT), $100,000$ steps (iNaturalist) or $10,000$ steps (K49-LT). We use Adam (Kingma & Ba, 2014) with learning rate 0.0001 and batch size 1.

## 5.2 RESULTS

**K49-LT.** In Section 4.2 we saw that using GIT to augment training led to more uniform invariance against each dataset's nuisance transform on K49-BG-LT and K49-DIL-LT, with the largest improvements on the smallest class sizes. Table 1 shows that GIT also improves balanced accuracy over DRS, for both the CE and LDAM losses. For K49-BG-LT, we find that GIT actually outper-

| Baseline | Strategy | Dataset | | | | | |
|---|---|---|---|---|---|---|---|
| | | K49-BG-LT | K49-DIL-LT | GTSRB-LT | CIFAR-10 LT | CIFAR-100 LT | TinyImageNet-LT |
| ERM | | $39.49 \pm 1.47$ | $39.49 \pm 1.47$ | $68.88 \pm 1.75$ | $70.74 \pm 0.13$ | $38.69 \pm 0.32$ | $16.33 \pm 0.10$ |
| CE + DRS | | $42.21 \pm 1.36$ | $39.48 \pm 1.47$ | $64.45 \pm 1.15$ | $74.28 \pm 0.56$ | $40.97 \pm 0.40$ | $16.98 \pm 0.18$ |
| | +GIT | $\mathbf{49.99 \pm 1.25}$ | $\mathbf{49.18 \pm 1.23}$ | $\mathbf{75.19 \pm 0.50}$ | $\mathbf{77.25 \pm 0.18}$ | $\mathbf{42.73 \pm 0.27}$ | $\mathbf{17.43 \pm 0.18}$ |
| | +Oracle | $44.22 \pm 1.36$ | $53.55 \pm 1.21$ | — | — | — | — |
| Focal + DRS | | — | — | $65.68 \pm 2.09$ | $73.51 \pm 0.50$ | $40.77 \pm 0.21$ | $16.87 \pm 0.17$ |
| | +GIT | — | — | $\mathbf{71.29 \pm 0.73}$ | $\mathbf{76.87 \pm 0.14}$ | $\mathbf{41.25 \pm 0.26}$ | $\mathbf{17.70 \pm 0.18}$ |
| LDAM + DRS | | $54.08 \pm 1.21$ | $50.44 \pm 1.24$ | $77.25 \pm 1.29$ | $76.73 \pm 0.74$ | $43.21 \pm 0.31$ | $20.54 \pm 0.21$ |
| | +GIT | $\mathbf{58.86 \pm 1.11}$ | $\mathbf{56.76 \pm 1.11}$ | $\mathbf{81.39 \pm 0.98}$ | $\mathbf{78.76 \pm 0.19}$ | $\mathbf{44.35 \pm 0.21}$ | $\mathbf{21.99 \pm 0.23}$ |
| | +Oracle | $54.49 \pm 1.12$ | $60.37 \pm 1.07$ | — | — | — | — |

Table 1: Average balanced test accuracy with or without GIT on different benchmarks. Adding GIT improves all resampling methods, with LDAM+DRS+GIT performing best overall. Uncertainty corresponds to 95% CI's across 30 instances for K49 variants, and the standard error over 3 training runs for the rest.

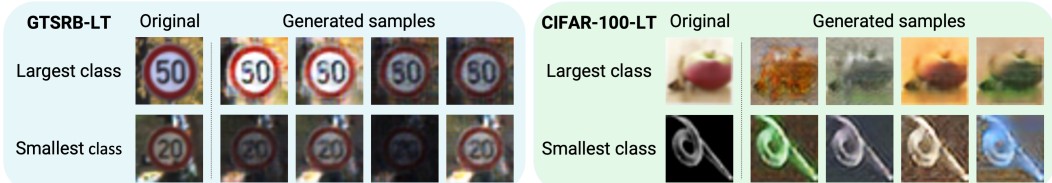

Figure 4: Samples from MIITNs trained to learn the nuisances of GTSRB-LT and CIFAR-100-LT, for use in GIT training. Each row shows sampled transormations of a single input image. We see the diversity of learned transformations, including changing lighting, object color/texture, and background.

forms the Oracle. This suggests that GIT may be learning natural nuisances from the original K49 dataset, in addition to the synthetic background intensity transformation.

**GTSRB-LT, CIFAR-LT, TinyImageNet-LT and iNaturalist.** Table 1 shows that adding GIT improves upon all three baseline methods on the GTSRB-LT, CIFAR-LT and TinyImageNet-LT benchmarks. Improvements are especially large on GTSRB-LT where street signs can appear under varying lighting conditions, weather, and backgrounds; here GIT improves LDAM+DRS by 4%. Figure 4 shows samples from the trained MIITNs: we see that it learns to vary lighting conditions, object color, and background, even for inputs from the smallest classes. Tables 3 and 4 show further combinations of GIT with different imbalanced training methods, where we see that GIT gives the largest improvements when combined with resampling based methods. Intuitively, GIT helps by augmenting examples from smaller classes, which appear more often when resampling.

In contrast to the previous benchmarks, adding GIT shows no improvements on iNaturalist (Appendix Table 5). Because iNaturalist is by far the largest of the datasets and contains the most classes, a more powerful MIITN may be needed to preserve class information while fully modeling the diverse nuisance transformations of this dataset.

## 6 ABLATION STUDIES

**GIT class size cutoff**: The samples generated by the trained MIITNs (Figures 2 and 4) capture a diverse set of nuisances, but have poorer image quality than the original images. When introducing GIT in Algorithm 1, we hypothesized that poor generated sample quality could hurt performance for classes that already have a lot of examples, so we introduced a class size cutoff $K$ to only augment classes with fewer than $K$ examples. By disabling this cutoff and applying GIT to all classes, we can inspect how GIT affects test accuracy for each individual class to validate this hypothesis.

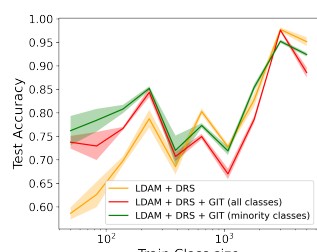

Figure 5: Test accuracy vs train class size for CIFAR-10 LT. Applied naively (in red), GIT performs better on smaller classes and worse on larger ones.

Figure 5 reports per-class performance for each of the ten classes in CIFAR-10-LT, arranged by class size. We see that using GIT without the cutoff boosts accuracy for the smaller classes, but can lower accuracy for the larger classes. In contrast, using the cutoff ($K = 500$) reduces the accuracy drop on the largest classes, while maintaining strong improvements on small classes. Table 2 directly compares **GIT** against the version with disabled cutoff, i.e. **GIT**

**(all classes)**. We see that using a class-size cutoff improves performance across the board. The differences are most pronounced for GTSRB followed by CIFAR-100 and CIFAR-10. We expect that this might be the case because CIFAR-10 has more coarse class definitions, such that the MIITN is less likely to corrupt the label.

**Automated data augmentation**: Since we can interpret GIT as a form of learned data augmentation, we also compare it to RandAugment (Cubuk et al., 2019), an automated data augmentation scheme carefully designed to have a small search space over augmentation parameters. As RandAugment has published tuned hyperparameters for CIFAR10/100, we use those and compare in CIFAR-LT. GIT outperforms RandAugment when using the LDAM loss, but RandAugment performs comparably to or better than GIT with CE or FOCAL loss. This indicates that RandAugment's set of transformations and search space work well in the CIFAR setting.

| Baseline | Strategy | Dataset | | |
|---|---|---|---|---|
| | | GTSRB-LT | CIFAR-10 LT | CIFAR-100 LT |
| CE + DRS | RandAugment | — | $\mathbf{78.78 \pm 0.99}$ | $\mathbf{45.53 \pm 0.20}$ |
| | GIT (all classes) | $66.22 \pm 1.12$ | $76.29 \pm 0.20$ | $39.77 \pm 0.16$ |
| | GIT | $\mathbf{75.19 \pm 0.50}$ | $\mathbf{77.25 \pm 0.18}$ | $42.73 \pm 0.27$ |
| FOCAL + DRS | RandAugment | — | $\mathbf{78.02 \pm 0.23}$ | $\mathbf{45.15 \pm 0.46}$ |
| | GIT (all classes) | $68.92 \pm 1.15$ | $76.44 \pm 0.34$ | $39.90 \pm 0.19$ |
| | GIT | $\mathbf{71.29 \pm 0.73}$ | $76.87 \pm 0.14$ | $41.25 \pm 0.26$ |
| LDAM + DRS | RandAugment | — | $77.67 \pm 0.07$ | $43.36 \pm 0.70$ |
| | GIT (all classes) | $78.51 \pm 1.29$ | $78.00 \pm 0.14$ | $43.24 \pm 0.22$ |
| | GIT | $\mathbf{81.39 \pm 0.98}$ | $\mathbf{78.76 \pm 0.19}$ | $\mathbf{44.35 \pm 0.21}$ |

Table 2: Ablations for the effects of the GIT class size cutoff, and RandAugment instead of GIT augmentation. We report the average balanced test accuracy and the standard error of the mean for 3 runs. **GIT** uses cutoff $K = 25, 500$, and $100$ for GTSRB-LT, CIFAR-10 LT and CIFAR-100 LT, respectively. This outperforms **GIT (all classes)** on all three datasets.

## 7    CONCLUSION

We study how deep neural network classifiers learn invariances to nuisance transformations in the class imbalanced setting. Even for simple simple class agnostic transforms such as rotation, we find that learned invariance depends heavily on class size, with the trained classifier being much less invariant for inputs from the smaller classes. This suggests that classifiers do not inherently do a good job of transferring invariances learned from the larger classes to the smaller classes, and is one way of explaining why they tend to perform poorly on heavily imbalanced or long-tailed classification problems. Motivated by the transfer problem, we explore Generative Invariance Transfer (GIT) as a method for directly learning a generative model of a dataset's nuisance transformations. We use this generative model during classifier training as a form of data augmentation to enforce invariance and observe both more uniform invariance learning across class sizes and improved balanced test accuracy on long-tailed benchmarks. Despite the observed improvements, GIT relies on being able to train a generative model of the dataset's nuisance transformations. The generative model may struggle to preserve class relevant information on large datasets with many classes. We speculate that future improvements in generative modeling could mitigate or resolve this issue. Since this work is about transferring invariances between classes, the focus is largely on class-agnostic transformations. Class-specific transformations would not be amenable to transfer, and handling them will likely require a different approach.

Our analysis also raises the question: *why* do deep classifiers struggle to transfer class-agnostic invariances across classes? We believe that an explanation for this effect is an interesting problem to be resolved by future work in the theory of deep learning. Additionally, although our analysis focused on the interaction between class size and invariance, these techniques used here could be extended to measure invariance in other contexts. Finally, a dataset can be class-balanced but imbalanced along other semantic dimensions not captured by the label, such groupings often studied in distributionally robust optimization (Sagawa et al., 2019). Further research on transferring invariances despite imbalance along more general dimensions could lead to more broadly generalizable and robust deep learning methods.

## 8 ACKNOWLEDGEMENTS

We would like to thank Archit Sharma and Eric Mitchell for insightful discussion during early phases of this work. We also gratefully acknowledge the support of Apple and Google.

## 9 REPRODUCIBILITY STATEMENT

For our baseline experiments, we use publicly available implementations and their existing hyperparameter values, linked or cited in Appendix B. For Generative Invariance Transfer we use a publicly available MUNIT implementation, with our modifications described in Appendix A. Classifier training with GIT is described in Algorithm 1. Since GIT effectively acts as a learned data augmentation and doesn't require modifying the classifier itself, for ease of comparison and reproducibility we kept details such as classifier architecture and optimization hyperparameters unchanged relative to the baselines. Appendix C.2 describes how to construct the K49-LT datasets, while Section 5.1 describe the GTSRB-LT and CIFAR-LT datasets. All code and pre-trained MIITN and classifier weights will be released upon publication.

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

## A    MUNIT TRAINING DETAILS

We implement our generative models using the MUNIT architecture and training algorithm (Huang et al., 2018), using the official source code at `https://github.com/NVlabs/MUNIT`. Although the architectures, optimizers, and most other training details are unchanged, MUNIT is designed to train on two datasets $A$ and $B$ and learns two generative models $G_{A \rightarrow B}$ and $G_{B \rightarrow A}$ to map between them. As we are only interested in learning transformations between images in the same (imbalanced) dataset, we set both $A$ and $B$ to be our one dataset and, after training, arbitrarily take $G_{A \rightarrow B}$ as the learned generative model $\tilde{T}$. Sampling $\tilde{x} \sim \tilde{T}(\cdot | x)$ corresponds to sampling different latent "style" vectors that are used as input to the MUNIT decoder. We observe in Figure 2 and Figure 4 that MUNIT produces a diverse set of transformed samples.

For all MUNIT training we use identical optimization hyperparameters as in the official source code (Adam (Kingma & Ba, 2014) with learning rate 0.0001 and batch size 1). We sample training examples with class balanced sampling, and as per the original implementation we rescale the pixel values between $[-1, 1]$ and apply random horizontal flips as data augmentation (images from iNaturalist-2018 is resized so that their shorter sides are of size 256, and then a random crop of $224 \times 224$ is taken from them without any padding). The image reconstruction loss has weight 10, while the adversarial loss, style reconstruction loss, and content reconstruction loss all have weight 1. We disable the domain-invariant perceptual loss. The generator architecture hyperparameters are identical to the "edges $\rightarrow$ handbags" experiment from Huang et al. (2018), and the discriminator is also the same except that we use a single scale discriminator instead of 3 scales. For K49-LT we train the MUNIT architectures on the $28 \times 28$ image inputs for $10,000$ steps. For GTSRB-LT and CIFAR-LT we train on $32 \times 32$ inputs for $140,000$ steps. For TinyImageNet-LT we train on $64 \times 64$ inputs for $200,000$ steps, and for iNaturalist-2018 we train on $224 \times 224$ inputs for $100,000$ steps.

## B    CLASSIFIER TRAINING DETAILS

**Optimization.** For fair comparison, nearly all of our classifier training hyperparameters are identical to those used in the CIFAR experiments of Cao et al. (2019): 200 training epochs with a batch size of 128, optimized by SGD with momentum 0.9 and with weight decay $2 \times 10^{-4}$. The initial learning rate is 0.1 and is decayed by a factor of 10 at 160 epochs and further decayed by the same factor at 180 epochs. Only the K49-LT experiments use a slightly modified training schedule of 50 epochs with learning rate decays at 30 and 40 epochs. For Delayed Resampling (DRS) the resampling stage starts at 160 epochs for GTSRB-LT and CIFAR-LT and at 30 epochs for K49-LT. For iNaturalist, following Cao et al. (2019), we train for 90 epochs with batch size 256 and learning rate 0.1, with further decaying the learning rate by 0.01 and 0.001 at epochs 30 and 60 respectively. The other hyper-parameters related to LDAM baseline are identical to (Cao et al., 2019).

**Data augmentation.** For the CIFAR-LT experiments we used random crop (with padding 4) and random horizontal flip (with probability 0.5) on all non-GIT methods. For methods that use GIT, random horizontal flip is first applied to the entire batch. Then we apply the GIT augmentation to half the batch, and the random crop augmentation to the other half. For TinyImageNet-LT, we use random crop to size $64 \times 64$ with padding 8. For iNaturalist, we first resize each training image to have the shorter side to have size 256. Next, we take a random crop of size $224 \times 224$ without any padding from this image or its random horizontal flip with equal probability. For all datasets, we normalize the images to the pixel mean and standard deviation of the respective training dataset.

Since MUNIT training uses random horizontal flip augmentation, we tried adding random horizontal flip augmentation for classifier training in K49-LT and GTSRB-LT experiments but found that it worsened performance on those datasets. This may be because handwritten characters and text on street signs are not actually flip invariant. Classifier training for K49-LT and GTSRB-LT does not use any additional data augmentation, except for the learned augmentations for the GIT methods.

**Architecture.** Like Cao et al. (2019), we use a family of ResNet (He et al., 2015) architecture implementations designed for CIFAR (Idelbayev, 2018) for all experiments. We use the ResNet20 architecture for the K49-LT experiments and the ResNet32 architecture for the GTSRB-LT and CIFAR-LT experiments. Finally, we use EfficientNet-b4 (Tan & Le, 2019) for TinyImageNet-LT and ResNet50 for iNaturalist.

## C    Full experimental details and results

### C.1    Methods

We use three loss functions for our experiments — the standard cross entropy loss (**CE**), **Focal** loss (Lin et al., 2017) and **LDAM** (Cao et al., 2019). The last two are specialized loss functions designed for imbalanced dataset learning. Following Cao et al. (2019), we choose $\gamma = 1$ as the hyper-parameter for Focal loss, and for LDAM loss, the largest enforced margin is chosen to be 0.5.

We couple these loss functions with a variety of training schedules, described below:

- Class balanced reweighting (**CB RW**) (Cui et al., 2019): Instead of reweighting the loss for a particular training example proportional to the inverse of corresponding class size, we reweight according to the inverse of the effective number of training samples in the corresponding class:

$$N_{eff}^i = \frac{1 - \beta^{N^i}}{1 - \beta} \tag{3}$$

    where $N_{eff}^i$ is the effective number of training examples in class $i$, $N^i$ is true number of training samples for class $i$, and $\beta$ is a parameter, which we typically set to 0.9999 for all our experiments.

- Class balanced resampling (**CB RS**) (Cui et al., 2019): We resample the training examples in a particular class with probability proportional to the inverse of effective number of training samples (see Equation 3) in that class.

- Resampling (**RS**): This is the regular resampling strategy, where we resample the training examples in a particular class with probability proportional to the inverse of (true) number of training samples in that class.

- Delayed reweighting (**DRW**) (Cao et al., 2019): We train in a regular way for the initial part of the training, and reweight the loss according to the inverse of the effective number of samples (Equation 3) in a training class only at the last phase of the training. For GTSRB-LT and CIFAR-LT, we typically train for 200 epochs, and reweight the loss function starting at 160 epochs.

- Delayed resampling (**DRS**): Similar to DRW, but we resample examples from class $i$ with a probability inverse to the **true** number of training examples in class $i$.

Among all the combinations of loss functions and training strategies, we discover that **LDAM + DRS** typically does the best, and combining GIT with it gives an additional boost in performance.

For GIT classifier training we set the class size cutoff parameter to $K = 25$ for GTSRB-LT, $K = 500$ for CIFAR-10-LT, $K - 100$ for CIFAR-100-LT and TinyImageNet-LT and $K = 20$ for iNaturalist.

### C.2    K49-LT

When constructing the synthetic datasets we used one of three transformation families and applied a randomly sampled transformation to each image in the dataset. For rotation, we rotated the raw image by a randomly sampled angle $\theta \in [0, 2\pi)$. For background intensity, we replaced the original black background with a randomly sampled pixel value between $0$ and $100$, where $255$ is the maximum intensity. For dilation and erosion, we randomly either applied dilation with $60\%$ probability or erosion with $40\%$ probability, using OpenCV's dilate and erode functionality. For dilation we used an $n \times n$ kernel of 1's, where $n \sim \text{Unif}(\{2, 3, 4\})$, and for erosion we used an $m \times m$ kernel of 1's, where $m \sim \text{Unif}(\{1, 2\})$.

The per class eKLD curves on the K49-LT variants were calculated for a ResNet20 architecture, but to verify that the results are not specific to ResNet we repeated these experiments on a simple CNN, which consists of four blocks of $3 \times 3$ convolutions, batch normalization, ReLU, and max pooling with stride 2. A final linear layer produces the output logits. Figure 6 shows the resulting per class eKLD for both architectures. We see that classifiers struggle to transfer learned invariances to the tail classes regardless of architecture, and that GIT can help mitigate this problem in both cases.

Figure 6: Per-class eKLD curves on three K49-LT variants for classifiers trained by different methods using two different architectures: ResNet20 (top) and a simple CNN (bottom). In each case wee see that classifiers learn invariance to the nuisance transform unevenly, with poor invariance for the tail classes. We also see that adding GIT helps reduce eKLD and improve invariance on the tail. Shaded regions indicate 95% CI's over 30 trained classifiers.

## C.3    GTSRB-LT AND CIFAR-LT

Table 3 and Table 4 present a more comprehensive set of experiments on GTSRB-LT and the two CIFAR-LT benchmarks, comparing how GIT performs with wider combinations of losses and training schedules. Overall we see that GIT works best when paired with a resampling method (such as DRS), and offers less benefit when applied to reweighting schemes or the standard training process. Intuitively, resampling boosts the probability of sampling examples from the small classes, which GIT can then augment, providing the classifier with more diversity for the small classes. Without resampling, the classifier is rarely sees examples from the small classes at all, minimizing the impact that GIT can have. In fact, without resampling, GIT will be applied primarily to images from the larger classes, and we saw in Section 6 that using GIT on larger classes can hurt the performance of the classifier. This is also confirmed on Table 3 and 4, where GIT often hurts the performance for non-resampling training schedules.

## C.4    INATURALIST-2018

We use the 2018 version of the iNaturalist dataset and report validation accuracy of different baselines in Table 5. Note that similar to Kang et al. (2020), we could not reproduce the numbers for various baselines from Cao et al. (2019).

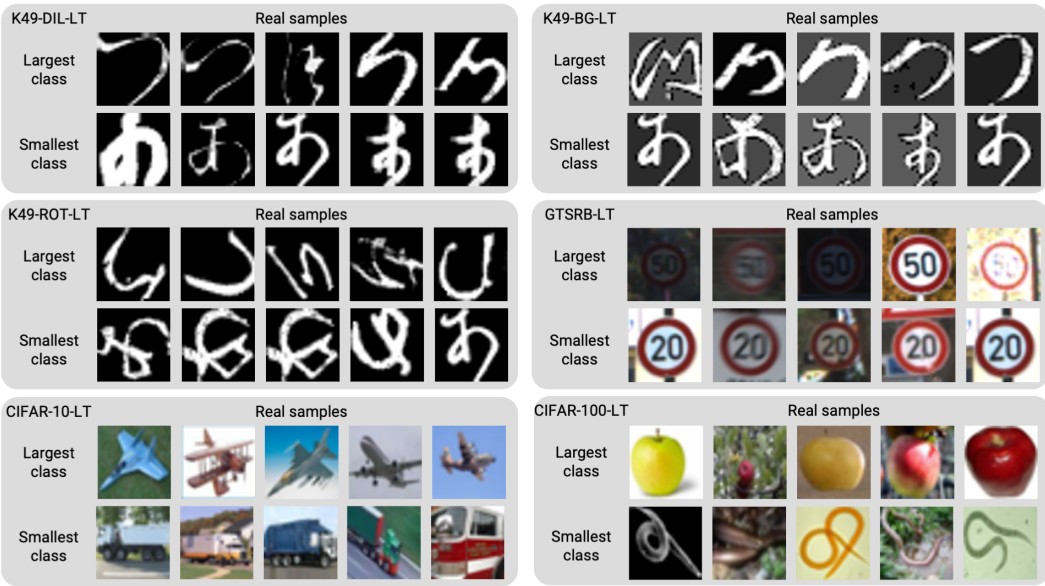

Figure 7: Random training examples from each of the 6 datasets considered in this work, with examples from both the largest and smallest classes in each dataset.

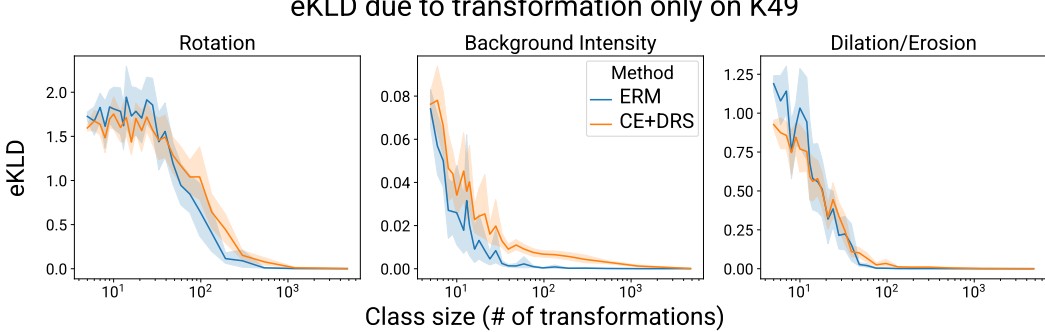

Figure 8: The expected KL divergence (eKLD) on synthetic K49 datasets that contain the same number of original K49 examples across all classes. These datasets are long-tailed only in the amount of transformations observed, and are designed to isolate the effect of the transformation. Each class starts with 5 examples from the *original* Kuzushiji-49 dataset. Larger classes are created by repeatedly sampling more transformations of the same 5 originals. This design ensures that the only difference between large and small classes is the amount of transformation diversity. These new datasets are easier to learn, but show the same qualitative trend as Fig. 1. Shaded regions show 95% CIs over 30 trained classifiers.

| Loss Type | Training Schedule | Without GIT | With GIT |
|---|---|---|---|
| CE | None | $68.88 \pm 1.75$ | $65.90 \pm 0.66$ |
| | CB RW | $41.31 \pm 6.45$ | $49.46 \pm 6.33$ |
| | CB RS | $52.36 \pm 4.55$ | $61.67 \pm 3.43$ |
| | RS | $55.06 \pm 1.28$ | $59.66 \pm 1.78$ |
| | DRW | $69.57 \pm 0.99$ | $71.30 \pm 1.17$ |
| | DRS | $64.45 \pm 1.15$ | $66.22 \pm 1.12$ |
| Focal | None | $69.07 \pm 1.14$ | $66.93 \pm 1.19$ |
| | CB RW | $38.42 \pm 9.61$ | $43.22 \pm 2.09$ |
| | CB RS | $55.30 \pm 0.04$ | $60.60 \pm 2.67$ |
| | RS | $57.74 \pm 4.60$ | $56.09 \pm 1.61$ |
| | DRW | $66.73 \pm 2.63$ | $66.80 \pm 0.99$ |
| | DRS | $65.68 \pm 2.09$ | $68.92 \pm 1.15$ |
| LDAM | None | $76.68 \pm 1.76$ | $76.87 \pm 0.61$ |
| | CB RW | $53.81 \pm 4.61$ | $59.69 \pm 2.92$ |
| | CB RS | $63.10 \pm 1.32$ | $74.58 \pm 0.44$ |
| | RS | $66.05 \pm 0.63$ | $73.61 \pm 1.22$ |
| | DRW | $76.41 \pm 1.26$ | $77.53 \pm 0.44$ |
| | DRS | $77.25 \pm 1.29$ | $\mathbf{78.51 \pm 1.29}$ |

Table 3: Full experimental results for the long-tailed GTSRB dataset. In this table, we report the average **balanced test accuracy** and the standard error of the mean of 3 runs for each entry. Note that we use GIT for all classes as opposed to only minority classes, in contrast to Table 1, but still we see improvements for most of the methods when combined with GIT. Section 6 shows that using GIT only for minority classes outperforms using GIT for all classes, but since that requires one more hyper-parameter to be tuned (class size threshold, K), we chose to use GIT for all classes in this table.

| Loss type | Training Schedule | Dataset | | | |
|---|---|---|---|---|---|
| | | CIFAR-10 LT | | CIFAR-100 LT | |
| | | Without GIT | With GIT | Without GIT | With GIT |
| CE | None | $70.74 \pm 0.13$ | $67.05 \pm 0.37$ | $38.69 \pm 0.32$ | $40.09 \pm 0.27$ |
| | CB RW | $71.90 \pm 0.14$ | $73.73 \pm 0.92$ | $30.06 \pm 0.61$ | $31.70 \pm 0.59$ |
| | CB RS | $69.85 \pm 0.08$ | $74.20 \pm 0.25$ | $32.13 \pm 0.79$ | $35.46 \pm 1.32$ |
| | RS | $69.15 \pm 0.62$ | $74.10 \pm 0.22$ | $33.08 \pm 0.21$ | $34.48 \pm 0.40$ |
| | DRW | $75.22 \pm 0.50$ | $75.77 \pm 0.09$ | $41.05 \pm 0.10$ | $41.90 \pm 0.32$ |
| | DRS | $74.28 \pm 0.56$ | $76.29 \pm 0.20$ | $40.97 \pm 0.40$ | $42.73 \pm 0.22$ |
| Focal | None | $70.22 \pm 0.56$ | $67.31 \pm 0.17$ | $38.41 \pm 0.27$ | $40.60 \pm 0.32$ |
| | CB RW | $69.22 \pm 0.70$ | $66.73 \pm 0.54$ | $26.80 \pm 0.85$ | $25.16 \pm 1.43$ |
| | CB RS | $68.64 \pm 0.62$ | $72.99 \pm 0.38$ | $33.04 \pm 0.36$ | $34.72 \pm 0.60$ |
| | RS | $69.30 \pm 0.20$ | $73.35 \pm 0.11$ | $32.73 \pm 1.13$ | $34.03 \pm 0.05$ |
| | DRW | $75.41 \pm 0.87$ | $74.94 \pm 0.41$ | $39.65 \pm 0.36$ | $40.14 \pm 0.16$ |
| | DRS | $73.51 \pm 0.50$ | $76.44 \pm 0.34$ | $40.77 \pm 0.21$ | $40.87 \pm 0.46$ |
| LDAM | None | $73.06 \pm 0.28$ | $69.00 \pm 0.38$ | $40.45 \pm 0.26$ | $40.88 \pm 0.73$ |
| | CB RW | $73.06 \pm 0.28$ | $69.00 \pm 0.38$ | $40.45 \pm 0.26$ | $40.88 \pm 0.73$ |
| | CB RS | $70.38 \pm 0.41$ | $75.46 \pm 0.07$ | $30.68 \pm 0.43$ | $32.37 \pm 0.39$ |
| | RS | $70.45 \pm 0.43$ | $75.02 \pm 0.04$ | $30.84 \pm 0.41$ | $33.50 \pm 1.01$ |
| | DRW | $77.13 \pm 0.27$ | $77.98 \pm 0.50$ | $43.11 \pm 0.33$ | $43.33 \pm 0.36$ |
| | DRS | $76.73 \pm 0.74$ | $\mathbf{78.00 \pm 0.14}$ | $43.21 \pm 0.31$ | $\mathbf{44.35 \pm 0.21}$ |

Table 4: Full experimental results for long-tailed CIFAR-10 and CIFAR-100 datasets. In this table, we report the average **balanced test accuracy** and the standard error of the mean of 3 runs for each entry. Bold numbers represent superior results for each dataset. Note that we augment all classes instead of only the minority classes for CIFAR-10 LT, and show that even without the hyper-parameter class size threshold, K, we see major improvements over most (Loss type, training schedule). For CIFAR-100 LT, using GIT to augment all classes often hurts the performance compared to not using GIT, so we only use GIT for classes with number of training samples $\leq 100$.

| Baseline | Strategy | Top-1 Accuracy | Top-5 Accuracy |
|---|---|---|---|
| ERM | | 47.265 | 72.063 |
| CE + DRS | | 57.574 | 78.707 |
| | +GIT | 54.986 | 76.578 |
| Focal + DRS | | 52.567 | 75.276 |
| | +GIT | 49.375 | 73.314 |
| LDAM + DRS | | 57.545 | 77.487 |
| | +GIT | 53.051 | 73.946 |

Table 5: Validation accuracy with or without GIT on iNaturalist-2018. We report the numbers from a single run due to long training times. We use GIT with class size cutoff $K = 20$.

