# OpenReview forum: "Do deep networks transfer invariances across classes?"
_ICLR.cc/2022/Conference — ICLR 2022 Poster_

### Official Review · Reviewer_4nAf · 2021-10-27

**Correctness:** 2
**Technical Novelty And Significance:** 2
**Empirical Novelty And Significance:** 3
**Recommendation:** 6
**Confidence:** 2

**Main Review:**

===================
Strengths:
+This paper provides a new perspective to understand the long-tailed problem of DNN models. They found that the class-agnostic invariances cannot automatically transfer from the head to tail classes, which means the models are simply memorizing the data without understanding it.  Since humans are able to disentangle the class-specific contents from class-agnostic transformations, they can easily learn new contents (categories) through a limited number of observations. Therefore, this paper pinpoints that the key to solving the long-tailed problem might be disentangling the class-specific features from class-shared features.

===================
Weaknesses:
- Despite the good motivation of this paper, the technical details of the paper are not satisfactory. For example, I think the authors confused the classifier with the entire model. A DNN classification model is usually composed of a backbone (like ResNet) and a classifier (e.g, Linear classifier or Cosine classifier). In algorithm 1, you only mentioned the classifier updating, but I guess it means the entire model here. Such a description is confusing in the study of long-tailed classification (LTC) because a commonly used strategy in LTC is called decoupling[1], which learns the backbone and the classifier separately.
- Second, according to Figure 3, the proposed method actually hurt the invariance of head categories while the CE+DRS doesn't, which means GIT is not a good method to transfer the learned class-agnostic invariances from head to tail classes. Unlike the head-tail trade-off in the accuracy, I don't think the learned invariances should be forgotten. The eKLD of the CE+DRS also proves that the trade-off shouldn't be existing in the given invariance metric.
- Besides, I don't think the GIT is necessary to increase the invariances of tail classes and the current ablation study is too simple. For example, how about using RandAug[2] for tail classes during training? It's much simpler than learning a generative model and I'm pretty sure that it can increase the invariance.

[1] Decoupling Representation and Classifier for Long-Tailed Recognition
[2] RandAugment: Practical automated data augmentation with a reduced search space

**Summary Of The Paper:**

=======================
Summary:
This paper studies the problem of how well do neural networks transfer class-agnostic invariances from head classes to tail classes. They found that even a given transformation is class-agnostic, the DNN models still cannot disentangle it from class-specific features, which could partially explain the worse performances on rare classes. To solve this problem, they introduce a GIT method that uses a generative model to augment the tail categories with more diverse samples under certain transformations. It can empirically improve the long-tailed performances and transfer the knowledge of class-agnostic transformations from head to tail.


**Summary Of The Review:**

===================
Justification:
This paper provides a new perspective to understand the long-tailed problem of DNN models, yet, the technical details and the solutions of the paper are not satisfactory

---

> ### Author Response · Authors · 2021-11-17
> **Further comparison with data augmentation**
>
> We thank the reviewer for their comments and suggestions.
>
> > A DNN classification model is usually composed of a backbone (like ResNet) and a classifier (e.g, Linear classifier or Cosine classifier). In algorithm 1, you only mentioned the classifier updating, but I guess it means the entire model here.
>
> We have updated this terminology in Algorithm 1 to clarify that the entire model is being updated.
>
> > how about using RandAug[2] for tail classes during training
>
> We repeated some of the CIFAR10/100 experiments for 3 seeds using RandAugment on top of the existing RandomCrop + RandomHorizontalFlip augmentation. We observed a slight benefit for adding RandAugment when training with LDAM+DRS (also including non-RandAug numbers for easy reference):
>
> |          | LDAM+DRS      | +GIT          | +RandAug      |
> |----------|---------------|---------------|---------------|
> | CIFAR10  | $76.73 \pm 0.74$ | $\mathbf{78.76 \pm 0.19}$ | $77.67 \pm 0.07$ |
> | CIFAR100 | $43.21 \pm 0.31$ | $\mathbf{44.35 \pm 0.21}$ | $43.36 \pm 0.70$ |
>
> We are still running the experiments for other combos besides LDAM + DRS, we will update the results (Tables 3 and 5) once those results are in.
> [Edit update]: We have finished running the RandAugment comparisons for the other loss functions (see Table 2). Interestingly, the results depend a lot on loss function. RandAug does better (or the same) as GIT for CE and Focal losses in CIFAR-LT, but worse with LDAM loss.
>
> > the proposed method actually hurt the invariance of head categories
>
> The visualization that you are referring to shows an intermediate/ablation version of the GIT algorithm that applies augmentation to all of the classes. This observation is exactly why the full GIT method applies augmentation only to the tail classes (see ablation in Section 6 which explores this further), so that imperfections in the generative model do not hurt invariance on the head classes. We will update Figure 3 to show both this ablation and the full version of the GIT method, with clarified names to reference each version.

---

> > ### Comment · Reviewer_4nAf · 2021-11-18
> > **Re: Further comparison with data augmentation**
> >
> > Thank you for your replies. I've raised my final rating from 3 to 5. based on your responses, but they only partially reassure my conerns. Without experiments on large-scale datasets like ImageNet-LT and comparisons with other stronger LT baselines, it's hard to evaluate the effectiveness of the proposed methods. Specifically, the improvement of RandAug on CIFAR-10/-100 might be limited, but these kinds of simple augmentations are much more effective and reliable in large-scale real-world images, while the generative model, on the other hand, usually performs worse in those datasets. Therefore, I can only raise my final score to 5.

---

> > > ### Author Response · Authors · 2021-11-26
> > > **Re: Further comparison with data augmentation**
> > >
> > > Thank you for your prompt reply! We would welcome further elaboration or references on the point that generative models would be worse on larger datasets. Other applications [1] of similar generative models have worked quite well on Imagenet. See also our recently added TinyImagenet-LT results (Table 1), where GIT again provides a benefit. We also note that our experimental settings (KMNIST, CIFAR10/100, TinyImageNet) are similar in scope to previous ICLR 2021 publications that leverage generative models for data augmentation [2,3].
> > >
> > > [1] Robey et al. Model-Based Robust Deep Learning: Generalizing to Natural, Out-of-Distribution Data.
> > >
> > > [2] Sinha et al. Negative Data Augmentation. ICLR 2021.
> > >
> > > [3] Goel et al. Model Patching: Closing the subgroup performance gap with data augmentation. ICLR 2021.

---

> > > > ### Comment · Reviewer_4nAf · 2021-12-05
> > > > **Re: Re: Further comparison with data augmentation**
> > > >
> > > > Sorry for the late reply. After thoroughly reading all the reviews especially the Official Review of Paper879 by Reviewer PgZ2, I decided to further raise my score to 6 to encourage the novel perspective of the long-tailed problem proposed by this paper.

---

> > > ### Author Response · Authors · 2021-11-30
> > > **Further questions/comments?**
> > >
> > > We hope our updates and comments have addressed your concerns. Just wanted to check whether you had any further questions or comments.

---

### Official Review · Reviewer_Zq9K · 2021-10-31

**Correctness:** 4
**Technical Novelty And Significance:** 2
**Empirical Novelty And Significance:** 3
**Recommendation:** 5
**Confidence:** 5

**Main Review:**

=== Strengnth ===
1. The paper is mostly well-motivated, well-written, and very easy to read.

2. The question raised by the authors in Section 1 is interesting. The experiments and metrics in Section 3 are inspiring and well-designed. They point out one direction for future research.

3. Overall, I think the direction the authors proposed to approach imbalanced classification is quite interesting. Generative models have not been widely used in this task. The authors also make "accurate" claims: the proposed method is mainly to deal with task/dataset-specific (but class-agnostic) variation.

=== Weakness ===
1. The technical part of the paper (GIT) is not well-written, described, and justified. The authors only simply mentioned that they apply MUNIT but didn't have more discussion or provide formulations. For example, why is MUNIT the appropriate method to be used? Why can we learn anything meaningful (i.e., the class-agnostic variance) by turning MUNIT to learn the mapping between the "same" domains? Are there related works that learn to change the input image (e.g., style transfer) and can be applied here? Finally, the authors should have mentioned in section 1 that the generative model is conditioned on the input image.

2. The experimental results are not enough. The authors only use small-scale datasets, but not large-scale ones like iNaturalist or mini/tiny-imagenet. The smaller improvement on CIFAR than on characters/traffic signs also raises a question --- could the proposed method be applied to natural, more complicated images/objects? Besides, the authors only compare to CB, LDAM, RS/RW, which are a bit outdated (though they are proposed in 2019). Finally, the proposed method seems to be not stable (according to Table 5).

3. As the authors argue that the proposed method is not merely data augmentation, I would like the authors to empirically compare it to existing sophisticated data augmentation methods, e.g., Zoph. et al. Rethinking Pre-training and Self-training, NeurIPS 2020.

4. From the paper description and the qualitative results (Fig 2 and Fig 4), it seems that the proposed method can only make the class-agnostic background, color, lighting, and dilation/erosion changes. However, in many large-scale datasets, there exists class-specific or semantically-related variation: for example, the appearance changes of cats might be similar to lions than to buses. Also, in many large-scale datasets, many tail classes seem to suffer from insufficient observations of their appearance changes (like from different viewpoints, insufficient instances (e.g., bag/airplane styles)). These can be found in Fig 7. It seems that the proposed method cannot effectively resolve these variances (see Fig 4).

5. Some highly relevant works are missing (not cited and discussed).

Transferring data variance from many-shot to few-shot data (e.g., to generate more data)

[a] B. Hariharan et al. Low-shot visual recognition by shrinking and hallucinating features. In ICCV, 2017

[b] X. Yin et al. Feature transfer learning for face recognition with under-represented data. In CVPR, 2019

[c] Y. Wang. Low-Shot Learning from Imaginary Data. In CVPR, 2018

Learning class-agnostic information

[d] Y. Yang. Rethinking the Value of Labels for Improving Class-Imbalanced Learning, NeurIPS, 2020

Style transfer or image-to-image translation should be discussed in related works.

Finally, based on the proposed method, I was wondering if the method proposed by Mariani et al. (2018) is applicable. If so, it should be compared.

=== Minor weakness/questions ===
1. The paper could cite and reference more existing works. For example, in Section 1, there are insufficient references (which dataset, which augmentation method?).

2. When DR or DS is used, do the authors only apply the proposed augmentation to the final learning steps or the whole learning steps? As the proposed method does not simply over-or under-sample the examples in the original dataset, I think it could be applied to the entire learning process.

**Summary Of The Paper:**

This paper works on long-tailed or class-imbalanced learning. The authors found that the learned classifier in such a setting cannot effectively transfer the class-agnostic (in)variance in a dataset from the head classes to the tail classes, which causes poor classification performance for the tail classes. The authors thus proposed to learn such class-agnostic (in)variance via a generative model, and then use it to augment the training data of minor classes. The experimental results on several small-scale datasets demonstrate the effectiveness of the proposed method in improving long-tailed or class-imbalanced classification.

**Summary Of The Review:**

Overall, I enjoy reading the paper as it is well-motivated and well-written. The experiments designed in section 3 are quite interesting. However, the proposed method in section 4 is not well-described and justified; the experimental comparison in section 5 is not sufficient. I thus give a score "5" for now. More accurately, my score is between "3" and "5".

---

> ### Author Response · Authors · 2021-11-17
> **GIT is stable in the expected scenarios (with resampling)**
>
> Thank you for the careful review. We have included the mentioned literature in the related work. We would like to emphasize that in addition to generative invariance transfer (GIT), a key contribution of our work is an analysis of invariances learning and transfer, which may help inform the future development of new methods beyond GIT.
>
> > I would like the authors to empirically compare it to existing sophisticated data augmentation
>
> We repeated some of the CIFAR10/100 experiments for 3 seeds using RandAugment [1] (suggested by R4) on top of the existing RandomCrop + RandomHorizontalFlip augmentation. We observed a slight benefit for adding RandAugment when training with LDAM+DRS (also including non-RandAug numbers for easy reference):
>
> |          | LDAM+DRS      | +GIT          | +RandAug      |
> |----------|---------------|---------------|---------------|
> | CIFAR10  | $76.73 \pm 0.74$ | $\mathbf{78.76 \pm 0.19}$ | $77.67 \pm 0.07$ |
> | CIFAR100 | $43.21 \pm 0.31$ | $\mathbf{44.35 \pm 0.21}$ | $43.36 \pm 0.70$ |
>
> We are still running the RandAug experiments for multiple seeds on other combos besides LDAM+DRS, we will update the results (Tables 3 and 5) once those results are ready.
>
> > Finally, the proposed method seems to be not stable (according to Table 5)
>
> Table 5 (now Table 4) shows many combinations of GIT with other imbalanced-class methods, and you are correct that GIT helps in most, but not all, combinations. However, the results are consistent and intuitive: GIT consistently helps any method that involves resampling (containing “RS” in the name) but may not help in situations without resampling. Without resampling, most of the GIT augmentation is applied to examples from the larger classes, whereas Section 3 suggests we should really be augmenting examples from the smaller classes. Resampling ensures that adequate GIT augmentation is applied to the smaller classes. In summary, GIT does not make sense as a method without resampling, which is exactly the scenario where it performs poorly. We have updated Section 5 to clarify this point.
>
> > it seems that the proposed method can only make the class-agnostic [...] changes. However, in many large-scale datasets, there exists class-specific or semantically-related variation [...]
>
> You are correct that the paper is largely focused on class-agnostic variation, whereas class-specific variation is largely beyond the scope of this work. This style of approach is naturally limited to transformations that can be transferred between classes, i.e. class-agnostic variation. We have updated the conclusion to discuss this as a natural limitation and area for future research. Note that we still observe improvements in natural datasets simply by improving transfer of class-agnostic invariances.
>
> [1] Cubuk et al. RandAugment: Practical automated data augmentation with a reduced search space. 2019.

---

> ### Author Response · Authors · 2021-11-18
> **More experiments**
>
> Some additional follow-up:
>
> > The authors only use small-scale datasets, but not large-scale ones like iNaturalist or mini/tiny-imagenet.
>
> As suggested, we are repeating our experiments on the TinyImageNet dataset. To make it long-tailed we follow the same procedure as for our CIFAR-LT experiments (also used in [1]): we remove training examples from each class to create an exponential imbalance with an imbalance ratio (between largest and smallest classes) of 100. We use a EfficientNet-b4 [2] backbone for our experiments, trained for 200 epochs with an SGD optimizer with momentum 0.9 and initial learning rate 0.1. We also use a cosine-annealing learning rate scheduler [3]. For GIT, we train a MUNIT for 200,000 steps with batch size 1 and Adam optimizer with initial learning rate 0.0001.
>
> These are the class-balanced test accuracies for LDAM + DRS with mean and standard error over 3 seeds:
>
> | LDAM+DRS      | +GIT          |
> |---------------|---------------|
> | $20.53 \pm 0.21$ | $\mathbf{21.99 \pm 0.23}$ |
>
> We see a similar improvement as in CIFAR-LT. GIT also helps when using CE or Focal loss (see Table 1).
>
> > The authors only simply mentioned that they apply MUNIT but didn't have more discussion or provide formulations.
>
> MUNIT is capable of learning a multimodal distribution of transformations of the input, which is good for capturing the full diversity of nuisance transformations available. You are correct that although MUNIT seems to work well for our cases, GIT is not exclusive to using MUNIT, and there are other approaches which could have their merits or drawbacks. We have updated Section 4.1 to discuss this and the Related Work now also discusses other style transfer and image to image translation approaches.
>
> [1] Cao et al. Learning Imbalanced Datasets with Label-Distribution-Aware Margin Loss. 2019.
>
> [2] Mingxing Tan, Quoc V. Le, EfficientNet: Rethinking Model Scaling for Convolutional Neural Network.
>
> [3] Ilya Loschilov, Frank Hutter, SGDR: Stochastic Gradient Descent with Warm Restarts.

---

> ### Author Response · Authors · 2021-11-27
> **Follow up questions/comments?**
>
> We hope we've addressed your concerns via the previous comments and paper updates. Please let us know if you have any follow-up questions or comments.

---

> > ### Comment · Reviewer_Zq9K · 2021-11-29
> > **Thank you for the rebuttal**
> >
> > I thank the authors for the rebuttal. The authors have tried to address many of my concerns. However, overall I still think the paper can be strengthened with:
> >
> > 1) A better organization/description of section 4. My concern on "The technical part of the paper (GIT) is not well-written, described, and justified" is still not fully addressed.
> >
> > 2) The numbers on Tiny-ImageNet are quite different from Cao et al. (NeurIPS 2019). It will definitely be great if the authors can include "iNaturalist", which has become a standard dataset on long-tailed classification.
> >
> > 3) More discussion on the limitation of the work. I do understand that class-specific variation is a bit beyond the scope of the paper, but I also wonder if it is indeed the major cause of variation for large-scale real images. The smaller improvement on CIFAR and the competitive performance by RandAug also makes me wonder if the authors can truly learn meaningful class-agnostic variations for CIFAR, Tiny-ImageNet, or iNaturalist.
> >
> > I will keep my score as "5".

---

> > > ### Author Response · Authors · 2021-11-30
> > > **Re: Thank you for the rebuttal**
> > >
> > > Thanks for the reply and explaining your reasoning, your points can definitely help us make the paper stronger.
> > >
> > > 1. Could you elaborate on what parts of the GIT description or justification could be further improved? We have updated it to discuss how we chose MUNIT since it can learn a diverse and multimodal space of transformations, and clarified that GIT is *not* MUNIT-exclusive (apologies for the typo in the current draft). And we discuss other methods for transforming images in the Related Work, which could be considered instead.
> > > 2. Thanks for pointing this out! We believe we have identified the source of differences on TinyImagenet, which is likely in the architecture and data augmentation. We will update the paper after removing those difference and re-running everything. Matching all details here has taken slightly longer since, unlike the CIFAR-LT experiments, there doesn't appear to be public code for the Cao et al TinyImagenet experiments. We will also repeat the experiments with iNaturalist and include those results.
> > > 3. RandAug itself is a carefully parameterized space over class-agnostic augmentations, which encodes a lot of human knowledge about natural variation in image classification. We believe that the results suggest GIT is capable of learning class-agnostic variation from data, to a level comparable to sophisticated human-designed augmentation schemes.

---

### Official Review · Reviewer_k4Fm · 2021-11-01

**Correctness:** 2
**Technical Novelty And Significance:** 3
**Empirical Novelty And Significance:** 3
**Recommendation:** 6
**Confidence:** 4

**Details Of Ethics Concerns:**

No concerns.

**Main Review:**

Strengths

+ Very interesting and fundamental research question
+ Insightful analysis
+ Well written, clear, and easy to understand
+ The GIT solution improves


Weaknesses

- I find the empirical risk minimization setting strange: the paper assumes a different distribution of the training data from the test data.
- I am not convinced of the main result in fig 1 (It could "just" be the class frequency; and not the transformation). See my detailed review below.
- The abstract claims an "explanation" but this is actually what I miss. Why is this effect is happening? Sure, a generative model can remedy the symptoms to some degree, but what is actually the cause of these symptions?
- Paper is not self-contained (I need the appendix for essential parts, thus, the appendix becomes part of the page limit?). See my detailed review below.
- I miss an analysis on GIS for "rotation" (fig 1 vs fig 3)
- I find it strange that the effect only seems to hold for supervised learning. So, if the class labels are removed but the dataset is the same, does the effect no longer occur? (ie: the GIS is trained unsupervised and seems to not have trouble in learning these general nuisance effects..)

Detailed review:

- Minor detail in general: The paper is fond of using "very" very much. A small tip is to try to avoid "very" it achieves the opposite of what the writing aims to achieve: http://www.writerswrite.co.za/45-ways-to-avoid-using-the-word-very/

- Minor detail: Abstract: "In order to" can nearly always be replaced by "To".

- Abstract: "much less invariant on smaller classes", what does this mean? Isn't invariance to a transformation T a binary property? ie: T(f(x)) = f(x) or is some (relative?) distance/similarity measure implied? I also assume its not about the classes, but about the nuisance transformation T in those classes? Perhaps the word "robust" is better?

Page 1: "where withholding explicit class information" so, is it then about the class information? Or about the occurrence frequency of the appearance of a class? Ie: this statement makes me wonder if class labels play a role here at all? Ie: in unsupervised settings there can also be an imbalance in the 'semantics' as measured by the nr of samples (but no explicit class labels are used). Basically: Why does this effect hold when using labels, and why does this effect not hold without using labels (but using the same dataset). It seems to me that the generative approach proposed later by this paper does not suffer from this effect?

Fig 1: How can we be sure that this effect is due only to the transformation and to nothing else than the transformation? I could pose an alternative explanation to this graph which is: Its generally more difficult to learn the classes that have fewer examples. This would also explain the shape of these curves, without relying on transformations. Ie: I miss a figure where there is no transformation at all, to see the baseline effect of having fewer examples (perhaps this can be achieved by measuring the eKLD for different initialized runs of the same model trained on untransformed data only? This would give an estimate of P_w || P_w ). I assume that this "no transformation baseline" would follow the same trend for ERM; yet perhaps be completely uniform for CE+DRS (?) So, generally, for this figure, I wonder how to investigate the exclusive effect of the transformation without measuring the confounded effect that "classes with fewer samples are more difficult".

Page 2, minor detail: "in long-tailed and class-imbalanced settings", Its a bit unclear to me what the difference between these two setting is (?).

Page 2: "we find that combining resampling methods with GIT further improves", it seems to me that GIT itself is another version of re-sampling? Perhaps a more specific method geared towards invariance?

Eq 1: Minor detail: shouldn't "x" be "x^{i}" ?

Page 3: "want our model to perform well across all classes". This is strange to me. In ERM a strong assumption is that P_train is distributed identically as P_test. In the setting where performing well over all classes is required, thus changes P_test from P_train. Does it make sense at all to use ERM then? Put in another way: if some classes are rare in the considered problem setting (which is why they are rare in P_train) then why is it to be expected that suddenly they do occur frequently when deploying the model? (not rare in P_test) ? Ie: optimizing *average* risk seems not the final goal here?

Page 4: About the conclusions about fig 1: see my earlier remark (how can we be sure that this effect is just because of the transformations, and not due to the small nr of samples?)

Page 5: I'm a bit confused what conclusion is now given by "Hence modeling the transformations directly is the more natural choice on imbalanced datasets". This paragraph seems to argue that a generative model does *not* suffer from class frequencies? Is this the goal of the paragraph? And if so, is that claim then true?

Page 6-7: All classifier/architecture information is missing. A paper should be self-contained. If it is mandatory to read the appendix to understand the paper then the appendix is an essential part of the paper and thus the paper is over the page limit.

Fig 3: Where is the "rotation" experiment from Fig 1? The paper is incomplete. (If the appendix contains such essential results, then, in my opinion, the appendix becomes part of the main submission and should adhere to the same page limits that hold for other authors)




**Summary Of The Paper:**

The paper investigates if robustness (or "invariance") to nuisance transformations which do not change the class label such as lighting, rotation etc are learned across all classes or if such robustness is sensitive to the class size. The paper demonstrates that such invariances seem not to transfer across classes: ie the classes with fewer examples suffer more. THe paper proposes a generative model (GIT) to "augment" the less frequent classes which, to some degree, remedies this problem.


**Summary Of The Review:**

I'm sorry if my review reads negative, but I really like the paper!  I think the question it asks is insightful, fundamental and important.
Yet, I am not fully convinced by how this question is then investigated (see my detailed review below).
I find the "GIS" part not so interesting, although it indeed remedies the symptoms and gives "bold numbers". I would be much more interested in *why* this happens. And if, for example, lighting plays a role, then why dont the first layers (which presumably are shared between classes) to some degree deal with this?
If the authors can somehow convince me, either with some new insights, or by new argumentation, I would be very happy to upgrade my score.

---

> ### Author Response · Authors · 2021-11-17
> **Updated to ensure the effect is from the transformation only**
>
> Thank you for your insightful and detailed comments. We have updated the paper to address minor writing issues and to make the paper more self contained (see blue colored text), and respond to other points here:
>
> > Fig 1: How can we be sure that this effect is due only to the transformation and to nothing else than the transformation? I could pose an alternative explanation to this graph which is: Its generally more difficult to learn the classes that have fewer examples. [...]
>
> You are correct that there are two sources of variation in the synthetic datasets: our synthetic transformation (e.g., rotation) and the natural variation of the original K49 dataset. Smaller classes can be more difficult because there are: (1) fewer examples from the original K49 dataset (2) fewer observed rotation angles. Mixing natural and synthetic variation is common for benchmarking invariance, e.g. RotMNIST [1,2,3], since a dataset with only a simple synthetic nuisance transformation is probably too easy.
>
> To answer the question and isolate the transformation effect we created new datasets where the amount of original K49 data is the same across all classes, and the only difference between large and small classes is the transformation diversity. We start with exactly 5 original K49 examples for each class. For the smallest class (size 5) we sample one transformation of each original, while for larger classes we repeatedly sample transformations of the 5 originals. We create the test examples of each class by sampling further transformations of the same 5 originals. This means the classifier does not need to generalize over natural variation, as the only difference between train and test examples of the same class is the transformation (e.g. they are at different rotation angles). This isolates the eKLD results to the transformation.
>
> We recalculated the eKLD for these new datasets in Section 3.2 and Figure 8. The qualitative effect is the same: invariance decreases with the “class size” (which now only corresponds to the number of transformations). Despite near-perfect invariance on the largest classes (note that these new datasets are easier to learn since we've removed almost all natural variation), this invariance is clearly not being transferred to the small classes.
>
> > In ERM a strong assumption is that P_train is distributed identically as P_test. [...] if some classes are rare in the considered problem setting then why is it to be expected that suddenly they do occur frequently when deploying the model?
>
> There are real world situations where we don’t want our model to reflect class imbalance in the training data. For example, in medical applications disease data is usually imbalanced but we still want to classify well on rare diseases [4]. In decision making we want to avoid discriminatory bias (i.e., to ensure fairness) [5]. Hence it is common in imbalanced classification literature to look at balanced (over classes) metrics [6,7], and we denote the “standard” training procedure “ERM'' following prior work [6]. We added this motivation to Section 3.1.
>
> > I miss an analysis on GIS for "rotation"
>
> We are not aware of any work training MUNIT-like models to learn rotation successfully. We have updated Section 4.1 and the conclusion to note this fact, and to emphasize that although the choice of generative model is somewhat arbitrary, GIT does rely on being able to learn a generative model of a dataset’s nuisance transformations. Note that we still observe improvements on GTSRB and CIFAR, so the ability to learn rotation seems to be not crucial for improvement on natural datasets.
>
> > So, if the class labels are removed but the dataset is the same, does the effect no longer occur?
>
> It is unclear how to evaluate invariance similar to the eKLD curves of Figure 1 without using class labels to train a model at some point, though we would welcome suggestions on this front. However, the intuition that the labels are partly responsible may correctly explain why training an unsupervised generative model can help, and we have updated our related work [8] (as mentioned by R3) to discuss this.
>
> [1] Larochelle et al. An empirical evaluation of deep architectures on problems with many factors of variation. 2007.
>
> [2] Finzi et al. Generalizing Convolutional Neural Networks for Equivariance to Lie Groups on Arbitrary Continuous Data. 2020.
>
> [3] Romero and Cordonnier. Group Equivariant Stand-Alone Self-Attention For Vision. 2020.
>
> [4] Bajwa et al. Computer-aided diagnosis of skin diseases using deep neural networks. 2020.
>
> [5] Hinnefeld et al. Evaluating Fairness Metrics in the Presence of Dataset Bias. 2018.
>
> [6] Cao et al. Learning Imbalanced Datasets with Label-Distribution-Aware Margin Loss. 2019.
>
> [7] Tang et al. Long-Tailed Classification by Keeping the Good and Removing the Bad Momentum Causal Effect. 2021.
>
> [8] Y. Yang. Rethinking the Value of Labels for Improving Class-Imbalanced Learning. 2020.

---

> ### Author Response · Authors · 2021-11-18
> **Additional explanations and the why question**
>
> > "Hence modeling the transformations directly is the more natural choice on imbalanced datasets". This paragraph seems to argue that a generative model does not suffer from class frequencies? Is this the goal of the paragraph? And if so, is that claim then true?
>
> The intuition we want to convey here is that an input-conditioned generative model of a class-agnostic transformation should transfer between classes, hence be less reliant on class size. Since the class-specific features are already present in the input (e.g., whether the subject is a cat or dog), the model only needs to learn the class-agnostic transformation (e.g., how to change the lighting of the input). By assumption, if the model can learn the class-agnostic transformation it should work well regardless of class (we verify this qualitatively for the MUNIT models in Figure 2--they work on both the largest and smallest classes). In contrast, a standard conditional GAN [1] receives a label and a noise vector as input, and must produce an image. This requires learning how to generate class-specific features.
>
> We have updated 4.1 to clarify this point.
>
> > Why is this effect happening?
>
> The purpose of Section 3 is to analyze how invariance learning relates to class size; the results suggest poor invariance transfer across classes since we see less invariance to the same transformation on smaller classes. This can explain poor generalization/balanced accuracy, but raises the new question: why do deep networks fail to transfer invariances between classes?
>
> That is not a question this work aims to answer. At the moment we are not aware of any theory of deep learning that is generally capable of explaining why transfer would or wouldn’t occur in a deep network, though we would welcome any relevant references. We have updated the conclusion to note this as an important and interesting area for future research as deep learning theory develops.
>
> [1] Mirza and Osindero. Conditional Generative Adversarial Nets

---

> ### Author Response · Authors · 2021-11-27
> **Follow up questions/comments?**
>
> We hope we've addressed your concerns via the previous comments and paper updates. Please let us know if you have any follow-up questions or comments.

---

> > ### Comment · Reviewer_k4Fm · 2021-11-28
> > **Thanks!**
> >
> > Thank you for your detailed answers and updates. Given this reply I will update my score with a +1.

---

### Official Review · Reviewer_PgZ2 · 2021-11-02

**Correctness:** 3
**Technical Novelty And Significance:** 3
**Empirical Novelty And Significance:** 4
**Recommendation:** 8
**Confidence:** 4

**Main Review:**

Strengths:

1. An extremely important problem: transferring invariances across classes opens up many avenues. For instance, to become invariant to any new transformation it would be sufficient to collect diverse data with only a few classes. This can help reduce resource required resources significantly.

2. The approach of modeling nuisance parameters using generative model and then propagating them to low frequency classes is very interesting.

3. The evaluation and experiment design is solid and rigorous.

4. Paper is very well written. It is easy to follow and understand.

Minor Weaknesses:

1. I am curious how GIT compares to other methods attempting to specifically enforce invariance. One strength of GIT is that it doesn't require knowing the transformation. However, in the case of dilation/erosion and background intensity showed here, the transformation is known. So, it is possible to get an exact value for an upper bound of what GIT could have achieved. It would be good to include this upper bound to know how well GIT performs.

2. Equation (1) is missing the superscript (i) for the image x.

3. Missing literature: Some recent literature on the role of dataset size/diversity on invariances is missing which would be good to comment and connect to [1,2].

4. I wonder where the most important contributions of the paper are. To me, it seems that bulk of the experiments are on quantifying their approach as a solution for class imbalance. So, I would suggest that the title and introduction should be adapted to focus more on this aspect. However, if instead the authors believe the main contribution to be the evaluation and solution for transfer of invariances, numbers mentioned in point 1 above should be mentioned.

References

1. Madan, S., Henry, T., Dozier, J., Ho, H., Bhandari, N., Sasaki, T., Durand, F., Pfister, H. and Boix, X., 2020. When and how do CNNs generalize to out-of-distribution category-viewpoint combinations? arXiv preprint arXiv:2007.08032.

2. Yang, G.R., Joglekar, M.R., Song, H.F., Newsome, W.T. and Wang, X.J., 2019. Task representations in neural networks trained to perform many cognitive tasks. Nature neuroscience, 22(2), pp.297-306

3. Achille, A. and Soatto, S., 2018. Emergence of invariance and disentanglement in deep representations. The Journal of Machine Learning Research, 19(1), pp.1947-1980.

**Summary Of The Paper:**

The paper investigates if invariances learned by the model transfer across classes. Focussing on class-agnostic nuisance parameters, the interplay between per-class size and invariant representations has been explored. It is suggested that while networks can become invariant to these parameters for classes with many examples, it is unclear if this is also the case for classes with fewer examples. The paper shows this is not true i.e. invariances do not transfer well to small classes, and suggest that improving this can help increase performance on imbalanced datasets. And so, the paper proposes a two step solution to this problem. First, an image-conditional generative model is learned which learns to transform the image such that only the nuisance parameter changes. Secondly, this model for nuisance parameters is used for data augmentation. Using this approach, the authors are able to achieve a significant improvement on standard long-tail datasets.

**Summary Of The Review:**

The paper investigates an extremely important problem: class-imbalance, and shows that part of the poor performance on classes with less samples stems from poor transfer of invariance to these classes despite good invariance building up for larger classes. Their proposed solution is thus very well motivated, and is a very nifty idea to model nuisance parameters. All in all, this is a solid applications paper in my opinion, and I think some minor changes in writing to focus more on fixing class-imbalance would make it a good paper for the computer vision community.

---

> ### Author Response · Authors · 2021-11-18
> **Thank you for your helpful comments and suggestions.**
>
> We have updated the related work to cite and discuss the suggested literature. (See blue text.)
>
> > I am curious how GIT compares to other methods attempting to specifically enforce invariance
>
> We are not aware of any general methods that can enforce invariance without additional assumptions such as labels, but we are running the suggested comparison in the synthetic K49 datasets since we have “oracle” access to the true nuisance transformations. We will update this comment and the paper once the results are available. [Edit]: These comparisons are now in Table 1 and Figure 3. The oracle augmentation works better than GIT for the dilation/erosion setting, unsurprisingly. Interestingly, GIT achieves higher accuracy than the oracle in the background intensity setting. This may be because GIT is also learning the natural nuisance factors of K49, which the oracle background intensity augmentation can't capture.
>
> Though they are far from “oracle” nuisance transformations, you might also be interested in the RandAugment results posted in another comment in this discussion.
>
> > I wonder where the most important contributions of the paper are. To me, it seems that bulk of the experiments are on quantifying their approach as a solution for class imbalance. So, I would suggest that the title and introduction should be adapted to focus more on this aspect. However, if instead the authors believe the main contribution to be the evaluation and solution for transfer of invariances, numbers mentioned in point 1 above should be mentioned.
>
> We view the main contribution of the paper to be the empirical study on the transfer of invariances across classes, since this study may inform the development of new methods, whereas GIT is just one possible approach for transferring invariances. While there are also a lot of experiments with GIT specifically, these experiments do provide further evidence for the hypothesis that invariances are not automatically transferred across classes and that explicit mechanisms for invariance transfer (such as those in GIT) can improve performance on imbalanced datasets.

---

### Decision · Program_Chairs · 2022-01-20

**Decision:**

Accept (Poster)

**Comment:**

This paper investigates how well properties invariant to changes such as lightening and background learned in the major class can be transferred to the minor class. In this paper, the authors reveal that invariances do not transfer well to small classes, and suggest that resolving this phenomenon can help increase the performance on imbalanced datasets. From this point of view, the authors propose a generative model-based augmentation technique.

Three reviewers suggested acceptance, and one reviewer judged borderline reject. It seems true that the method is not novel enough, but it is solid and well motivated. In particular, the finding of the paper is interesting and the design of the experiment is well done, so I think that it will have a great influence on research in this field in the future. As the negative reviewer mentioned, the lack of large-scale experiments is a major weakness of this paper. I strongly encourage the final version to supplement the promises made to the reviewer, including adding iNaturalist experiments.